# An unconventional cancer-promoting function of methamphetamine in hepatocellular carcinoma

Zizhen Si[1,2,*], GuanJun Yang[3,*], Xidi Wang[2], Zhaoying Yu[4], Qian Pang[2], Shuangshuang Zhang[2], Liyin Qian[2], Yuer Ruan[4], Jing Huang[1], Liu Yu[2],

**For the past decade, the prevalence and mortality of methamphetamine (METH) use have doubled, suggesting that METH use could be the next substance use crisis worldwide. Ingested METH is transformed into other products in the liver, a major metabolic organ. Studies have revealed that METH causes deleterious inflammatory response, oxidative stress, and extensive DNA damage. These pathological damages are driving factors of hepatocellular carcinoma (HCC). Nonetheless, the potential role of METH in HCC and the underlying mechanisms remain unknown. Herein, we found a higher HCC incidence in METH abusers. METH promoted cellular proliferation, migration, and invasion in two human-derived HCC cells. Consistently, METH uptake promoted HCC progression in a xenograft mouse model. Mechanistically, METH exposure induced ROS production, which activated the Ras/MEK/ERK signaling pathway. Clearance of ROS by NAC suppressed METH-induced activation of Ras/ERK1/2 pathways, leading to arrest of HCC xenograft formation in nude mice. To the best of our knowledge, this is the first study to substantiate that METH promotes HCC progression and inhibition of ROS may reverse this process.**

## Introduction

Methamphetamine (METH) is a highly potent amphetamine derivative that significantly affects psychiatric and physical output. METH is widely used for increasing physical activity, wakefulness, and decreasing appetite, and this widespread misuse has been associated with the intense euphoria METH produces (1). Reports suggest that METH use has increased worldwide, indicating that it is developing into its own epidemic (2).

It has been established that METH is a fatal and toxic chemical substance with multiple organ toxicities, including heart (3), brain (4) and other organs. The associated neurotoxicity and cardiotoxicity have been widely studied for years. METH is absorbed in the lungs and metabolized in the liver, where it is transformed into

circulating metabolites amphetamine and p-OHMA via the polymorphic cytochrome P450 2D6 enzyme (5). Studies have reported that METH and its metabolites can induce an aberrant inflammatory response, oxidative stress, and significant DNA damage (6, 7, 8), which are correlated with severe liver pathological changes and might eventually lead to hepatic failure or cancer (9, 10). However, little is currently known about the adverse effects of METH in liver cancer, especially in hepatocellular carcinoma (HCC).

Oxidative stress, caused by an imbalance between oxidative species production and antioxidant molecules in cells, plays a critical role in liver carcinogenesis and progression (11). It is widely acknowledged that reactive oxygen species (ROS) are the most abundant among all reactive species. The presence of elevated ROS levels promotes HCC through diverse signaling pathways, including renin–angiotensin system (Ras)/extracellular-regulated kinases 1 and 2 (ERK1/2) signaling pathways (12, 13, 14). Many studies substantiated that dopamine-dependent ROS production and oxidative stress induce METH toxicity with decreased glutathione levels, reduced levels and activities of antioxidant enzymes, and increased lipid peroxidation (15, 16, 17, 18). Accordingly, we hypothesize that METH leads to HCC through ROS-mediated Ras activation.

The present study showed that METH abusers and rehabilitees have a significantly higher HCC risk than normal subjects. METH facilitated cellular proliferation, migration, and invasion of HCC cells. Furthermore, METH uptake promoted HCC progression in a xenograft model. Next, we corroborated that the ROS-regulated Ras/MEK/ERK signaling pathway was involved in this process. Clearance of ROS inhibited METH-induced Ras activation and suppressed HCC progression. Overall, our research provided compelling evidence of the tumorigenic role of METH in HCC.

## Results

### METH use is associated with a higher incidence of liver cancer

The total sample consisted of 425 participants: 207 individuals were excluded, and 218 individuals completed the study (Fig 1). A total of

---

[1]Department of Pharmacy, The Affiliated Hospital of Ningbo University Medical School, Ningbo, P. R. China   [2]School of Medicine, Ningbo University, Ningbo, P. R. China   [3]State Key Laboratory for Managing Biotic and Chemical Threats to the Quality and Safety of Agro-Products, Ningbo University, Ningbo, P. R. China   [4]Department of Psychology, College of Teacher Education, Ningbo University, Ningbo, China

Correspondence: Liuyu@nbu.edu.cn
*Zizhen Si and GuanJun Yang contributed equally to this work

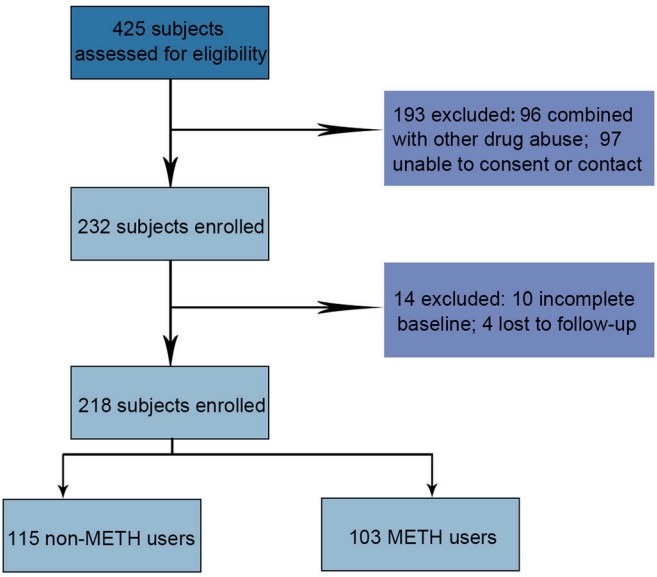

**Figure 1.** Diagram showing criteria to include and exclude participants.

425 subjects assessed for eligibility

193 excluded: 96 combined with other drug abuse; 97 unable to consent or contact

232 subjects enrolled

14 excluded: 10 incomplete baseline; 4 lost to follow-up

218 subjects enrolled

115 non-METH users

103 METH users

218 participants were included. Among these, 103 were diagnosed as METH users or METH rehabilitees (Table S1), and 115 were age- and gender-matched normal subjects with no history of METH abuse (Table S2). The demographic characteristics of the participants are shown in Table 1. There was a near-significant relationship between hepatitis B/C virus and METH use ($P$ = 0.058/$P$ = 0.053). The alcohol use was higher in the METH group ($P$ = 0.032). Furthermore, METH use was significantly associated with a higher incidence of liver cancer ($P$ = 0.004), suggesting that METH may play a critical role in HCC. As alcohol use and older age are highly risk factors for liver cancer, the incidence of liver cancer in METH users is separated by age or years of alcohol use. These results showed that neither alcohol use nor older age is associated with a higher incidence of liver cancer in METH users (Table 2), indicating that METH use is an independent risk factor for liver cancer.

## METH promotes cancer-associated properties in human liver cancer cells

To investigate the biological effects of METH in HCC, HUH7 and HepG2 cells were exposed to different concentrations of METH for 0, 24, 48, and 72 h. Then, the cell viability was determined by MTT assays and cell proliferation ability was determined by BrdU assays. The MTT assay showed that treatment with low concentrations of METH significantly promoted cell viability in HUH7 cells (0.1, 1, or 10 nM) and HepG2 cells (1, 10, or 100 nM) (Fig 2A and B). Then, HUH7 or HepG2 cells were exposed to METH for 72 h and underwent the BrdU assay. As shown in Fig 2C and D, low concentrations of METH promoted cell proliferation in HUH7 cells (0.1, 1, or 10 nM) and HepG2 cells (1, 10, or 100 nM) (Figs 2C and D and S1A and B). In addition, cell apoptosis was determined by Western blot and METH decreased the expression of cleaved caspase-3 (Fig S1C and D). Consistently,

**Table 1.** Distribution and characteristics of the sample.

| Variables | Frequency (percentages), means ± SD | | | *P*-value |
| --- | --- | --- | --- | --- |
| | Total sample (n = 218) | Non-METH users (n = 115) | METH use (n = 103) | |
| Sex | | | | |
| Male | 112 (51.38%) | 43 (38.39%) | 69 (61.61%) | 0.156 |
| Female | 106 (48.62%) | 72 (67.92%) | 34(32.08%) | |
| Age (yr) | | | | |
| 30–39 | 92 (42.20%) | 51 (44.35%) | 41 (39.81%) | |
| 40–49 | 83 (38.07%) | 43 (37.39%) | 40 (38.84%) | 0.236 |
| 50–60 | 43 (19.73%) | 21 (18.26%) | 22 (21.35%) | |
| HBV positive | | | | |
| Yes | 14 (6.42%) | 6 (5.22%) | 8 (7.77%) | 0.058 |
| No | 204 (93.58%) | 109 (94.78%) | 95 (92.23%) | |
| HCV positive | | | | |
| Yes | 7 (3.21%) | 3 (2.61%) | 4 (3.88%) | 0.053 |
| No | 211 (96.79%) | 112 (97.39%) | 99 (96.12%) | |
| Alcohol use | | | | |
| Yes | 147 (67.43%) | 62 (53.91%) | 85 (82.52%) | 0.032* |
| No | 71 (32.57%) | 53 (46.09%) | 18(17.48%) | |
| Liver cancer | | | | |
| Yes | 8 (3.67%) | 2 (25%) | 6 (75%) | 0.004** |
| No | 210 (96.33%) | 113 (53.8%) | 97 (46.2%) | |

HBV, hepatitis B virus; HCV, hepatitis C virus.

**Table 2.  Risk factors indicated in the METH users with liver cancer.**

| Variables | Liver cancer incidence in METH users (n, %) | P-value |
|---|---|---|
| Age (yr) | | |
| 30–39 | 1 (0.97%) | |
| 40–49 | 4 (3.88%) | 0.718 |
| 50–60 | 1 (0.97%) | |
| Alcohol use (yr) | | |
| 0–9 | 2 (1.94%) | |
| 10–19 | 3 (2.91%) | 0.639 |
| 20–30 | 1 (0.97%) | |

the colony formation assay showed that HUH7 cells and HepG2 cells treated with low concentrations of METH could form more colonies compared with the ddH2O control group (Figs 2E and F and S2A–D). We repeated some key experiments with poorly differentiated HCC cell lines corresponding to the late HCC stage (using the HLE cell line),

and the results were similar to those in HUH7 cells and HepG2 cells (Fig S3). However, HUH7 and HepG2 cells treated with high METH concentration did not exhibit higher proliferation potential (Fig 2A–F).

Next, the invasion and migration ability of the two types of cells (with 10 μg/ml mitomycin C) were determined by the transwell assay with or without Matrigel. Similar to the effects of METH on proliferation potential, low concentrations of METH treatment promoted the migration and invasion ability of HUH7 and HepG2 cells (Fig 2G–J). Furthermore, the wound healing assay also showed the same result (Fig S4A–D). Our results showed that 1 or 10 nM METH promoted the proliferation, migration, and invasion ability of HUH7 and HepG2 under all predefined concentrations of METH in this study.

## METH exposure activates Ras/ERK pathways in human liver cancer cells

Given that the Ras/ERK pathway has been shown to increase cell proliferation, we sought to determine whether METH could up-regulate the expression of Ras in HUH7 or HepG2 cells after

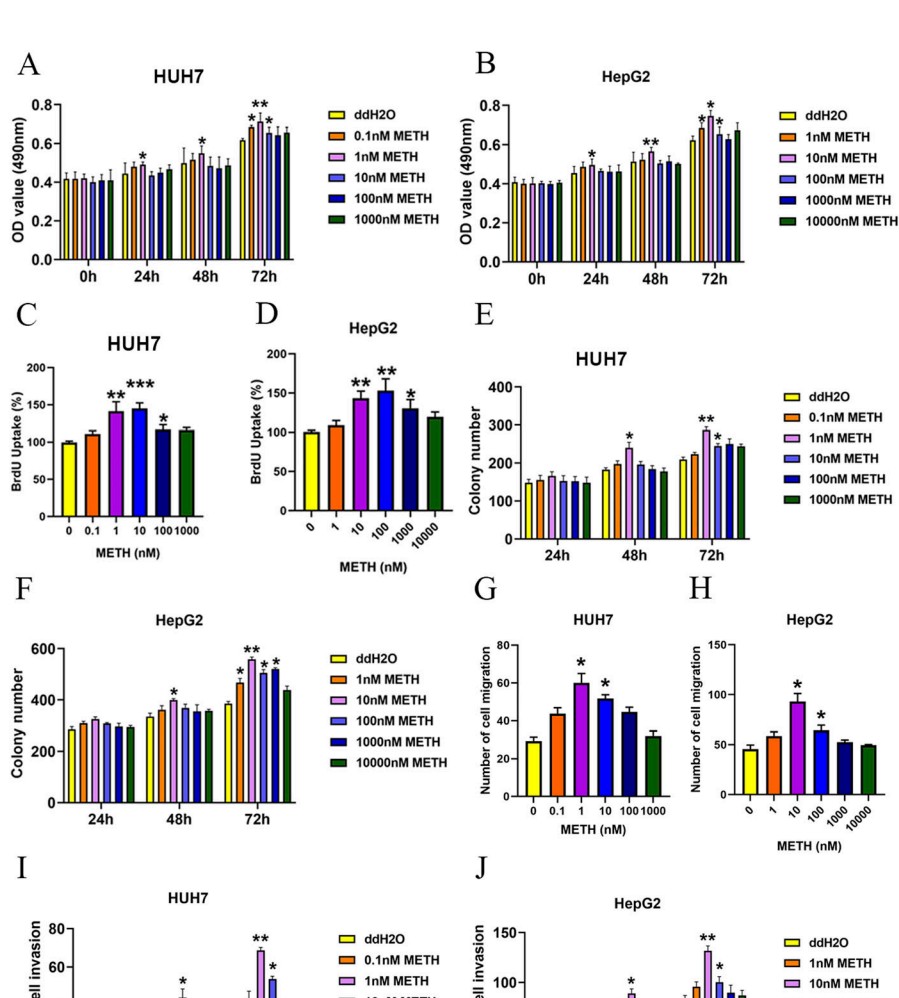

**Figure 2.  Impact of METH on HUH7 and HepG2 cells.**
**(A, B)** MTT assay was applied to determine the cell viability of HUH7 and HepG2 cells under various concentrations of METH. **(C, D)** Growth rate of HUH7 and HepG2 cells was determined by the BrdU assay. The BrdU value in untreated cells was set as 100%. **(E, F)** HUH7 and HepG2 cells were subjected to the colony formation assay. **(G, H, I, J)** Effect of METH on hepatocellular carcinoma cell migration and invasion was assessed with the transwell migration assay and the transwell invasion assay, respectively (mean ± SD of three independent experiments). *P < 0.05 and **P < 0.01.

exposure to METH. The Western blot results showed that after 48 or 72 h of METH treatment, the Ras expression increased significantly in HUH7 cells treated with 0.1, 1, and 10 nM METH (Fig 3A–D) and HepG2 cells treated with 1, 10, and 100 nM METH (Fig 3E–H). Furthermore, Ras activation was increased in HUH7 cells treated with 1 nM METH (Fig 3I and J) and HepG2 cells treated with 10 nM METH (Fig 3K and L). We further explored the downstream components of the Ras pathway and assessed the activation of p-ERK and p-MEK in HUH7 or HepG2 cells with METH exposure for 72 h. We found that METH increased the expression of p-ERK and p-MEK in HUH7 cells at concentrations of 1 or 10 nM (Fig 4A–D) and in HepG2 cells at 10 or 100 nM (Fig 4E–H). Furthermore, the expression of PI3K and MAPK was tested by Western blot and the results showed that METH increased the expression of PI3K and MAPK in HUH7 and HepG2 cells (Fig 4I–L). These results demonstrated that METH exposure activated the Ras/MEK/ERK pathway in HCC.

### ROS regulates Ras/ERK pathways in human liver cancer cells with METH exposure

To determine the upstream events leading to Ras up-regulation, we quantified the ROS levels in HUH7 cells and HepG2 cells treated with METH. The results showed that ROS levels were elevated under METH exposure (Fig 5A–D). NAC is an efficient scavenger of ROS. As expected, cotreatment with NAC reversed Ras expression and downstream p-ERK and p-MEK expression (Fig 5E and F) and further inhibited HepG2 cell proliferation, invasion, and migration (Fig 5G–I). Our results suggested that ROS was involved in METH-induced Ras/MEK/ERK pathways and regulated liver cell proliferation, invasion, and migration.

### METH exposure promotes xenograft tumor formation through ROS-induced Ras activation in vivo

To validate the influence of METH on liver cancer progression in vivo, nude mice were subcutaneously injected with HepG2 cells and treated with different concentrations of METH (0, 0.05, 0.5, 5, 50, and 500 μg/kg) by intraperitoneal injection every day. Strikingly, 0.5 or 5 μg/kg treatment of METH promoted the growth of HepG2 xenograft tumors (Fig 6A–C), which is consistent with the in vitro observations. Furthermore, the ROS levels were elevated in HepG2 xenograft tumors (Fig 6D and E). Taken together, these data confirmed that METH could significantly arrest the growth of xenograft tumors.

To further confirm the role of ROS in liver tumorigenesis under METH exposure, nude mice were subcutaneously injected with HepG2 cells and treated with 100 mg/kg of NAC by intraperitoneal injection every day. The results showed that the inhibition of ROS almost completely inhibited the growth of METH/HepG2 xenograft tumors (Fig 6F and G). What's more, NAC treatment reversed the levels of ROS and activation of Ras caused by METH in METH/HepG2 xenograft tumors (Figs 6D and E and 7A–D). In addition, Ras/MEK/ERK pathways caused by METH in METH/HepG2 xenograft tumors were also reversed by NAC treatment (Fig 7E and F). Together, these data confirmed that METH could promote the growth of HCC through ROS-dependent Ras/MEK/ERK pathways (Fig 8).

## Discussion

METH use has become a growing worldwide phenomenon, irrespective of wealth status, culture, and geographical location (19, 20). METH can have adverse and potentially fatal effects on the human body, including deleterious consequences to the neurological system, elevated blood pressure, atherosclerotic cardiovascular disease, and acute vasospasm (21).

Herein, we first substantiated that the incidence of liver cancer is significantly higher in the METH users compared to individuals with no prior drug exposure. The in vitro assay demonstrated that METH promoted cancer-associated properties such as cell proliferation, migration, and invasion ability, which was further confirmed by nude mouse experiments, consistent with a study by Ropek et al, who found that METH could induce DNA damage and micronucleus formation, which may ultimately cause cancer (22). These findings provide compelling evidence that METH promotes liver cancer progression, emphasizing that METH users and even METH rehabilitees are at high risk of liver cancer in addition to METH-associated neurotoxicity and cardiotoxicity.

An increasing body of evidence suggests that METH can cause toxicity by inducing ROS production and redox imbalance (23, 24). Besides, ROS induction has been well documented in multiple cancers and may lead to the activation of pro-tumorigenic signaling and induction of genetic instability or DNA damage (25, 26). It has been shown that elevated ROS levels can promote cell growth, proliferation, and survival by activating Ras/MEK/ERK signaling pathways (27, 28). In the present study, we found that METH treatment significantly increased ROS levels in liver cancer cells and xenograft tumors. Moreover, METH treatment up-regulated RAS expression and activated the MEK/ERK signaling pathway. Clearance of ROS by NAC suppressed METH-induced Ras up-regulation and MEK/ERK activation and further reversed METH-induced liver cancer progression. Accordingly, it is highly conceivable that METH could increase ROS levels and subsequently induce Ras expression, to activate the MEK/ERK signaling pathway in liver cancer cells.

Importantly, it should also be borne in mind that NAC could also be used against the progression and migration of cancer, irrespective of METH use (29, 30). It remains unclear whether NAC alone can affect these cancer cells. In any case, the reported beneficial effects of NAC provide a wide range of possibilities for developing novel therapeutic strategies against METH-induced liver cancer, for which few options are currently available.

In summary, our study revealed that METH use is associated with an increased risk of liver cancer. METH promotes liver cancer progression by ROS induction–mediated Ras up-regulation, which activates the MEK/ERK signaling pathway, whereas NAC reverses these effects via ROS clearance (Fig 8). These findings provide novel insights into our understanding of the harmful effects of METH other than neurotoxicity and cardiotoxicity.

## Conclusions

In this study, we found that METH use was associated with a higher incidence of HCC in METH abusers and rehabilitees. METH treatment significantly promoted cellular proliferation, migration, and

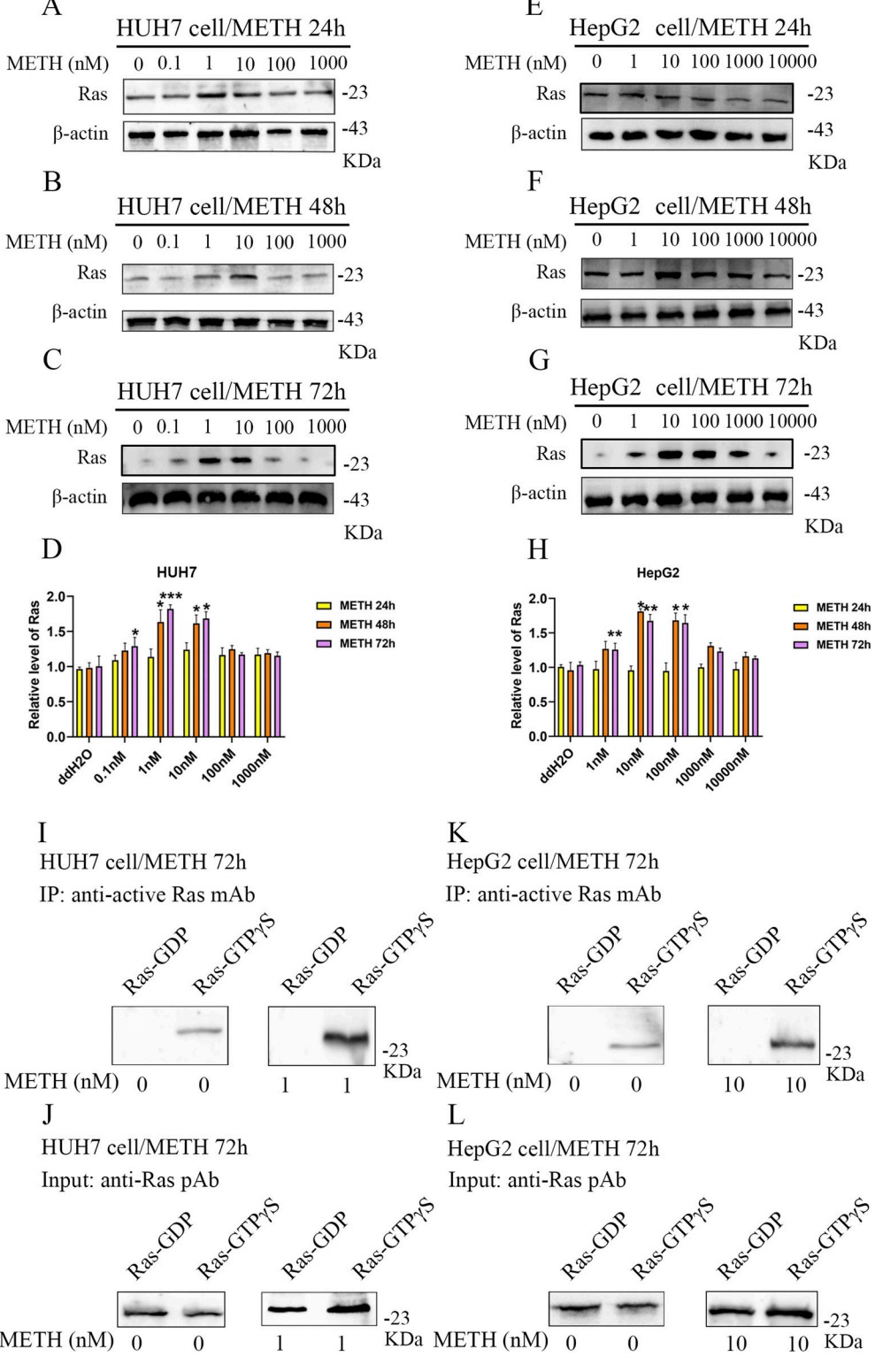

**Figure 3. Expression and activation levels of Ras were elevated with METH treatment.**
**(A, B, C, D, E, F, G, H)** Different concentrations of METH for 24, 48, and 72 h were applied to HUH7 and HepG2 cells, and Western blot was used to detect the expression of Ras (A, B, C, D, E, F, G, H) (mean ± SD of three independent experiments). $*P < 0.05$ and $**P < 0.01$. **(I, J, K, L)** Active Ras Pull-Down and Detection Kit was used to detect the activation of Ras in HUH7 (METH/1 nM/72 h) and HepG2 (METH/10 nM/72 h) cells.

invasion in two different human-derived HCC cells and in a xenograft model. Mechanistically, METH exposure induced ROS production, which activated the Ras/MEK/ERK signaling pathway. ROS scavenging by NAC abolished the METH-induced activation of Ras/ERK1/2 pathways leading to arrested HCC xenograft formation in nude mice. To the best of our knowledge, this is the first study to

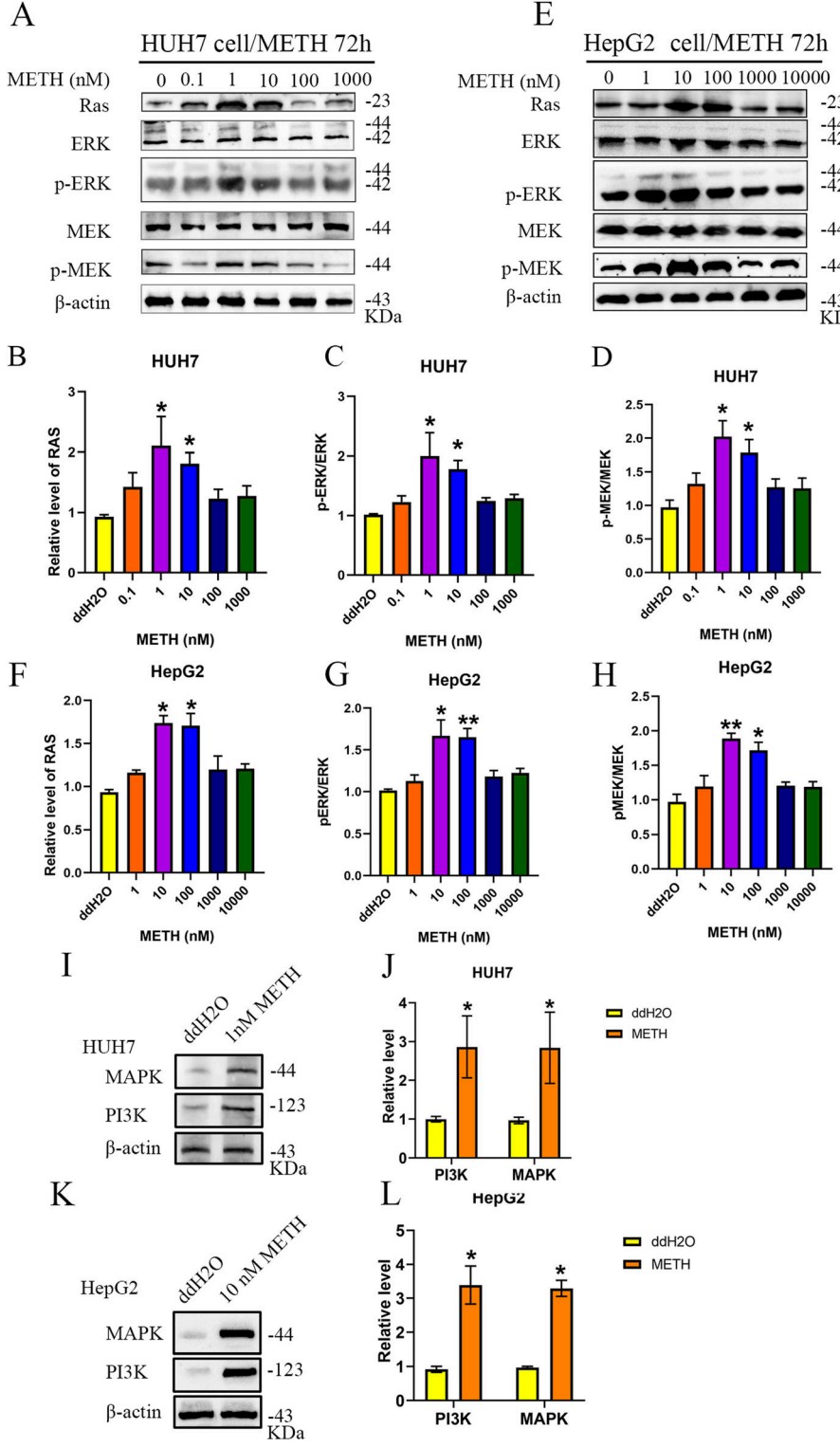

**Figure 4.   METH exposure up-regulated the protein level of the RAS/MEK/ERK pathway.**
HUH7 and HepG2 cells were exposed to different concentrations of METH for 72 h. **(A, B, C, D, E, F, G, H)** Western blot was used to detect the protein levels of Ras, p-MEK, MEK, p-ERK, and ERK in HUH7 (A, B, C, D) and HepG2 (E, F, G, H) cells. **(I, J)** Western blot was used to detect the protein levels of PI3K and MAPK under 1 nM METH exposure for 72 h in HUH7 cells. **(K, L)** Western blot was used to detect the protein levels of PI3K and MAPK under 10 nM METH exposure for 72 h in HepG2 cells (mean ± SD of three independent experiments). *$P < 0.05$ and **$P < 0.01$.

provide evidence of the close correlation between METH and HCC. We uncovered that the cancer-promoting function of METH in HCC was mediated by ROS-mediated Ras/MEK/ERK signaling pathway activation. Our study highlights the need to screen METH users and even METH rehabilitees who may be at high risk of liver cancer development.

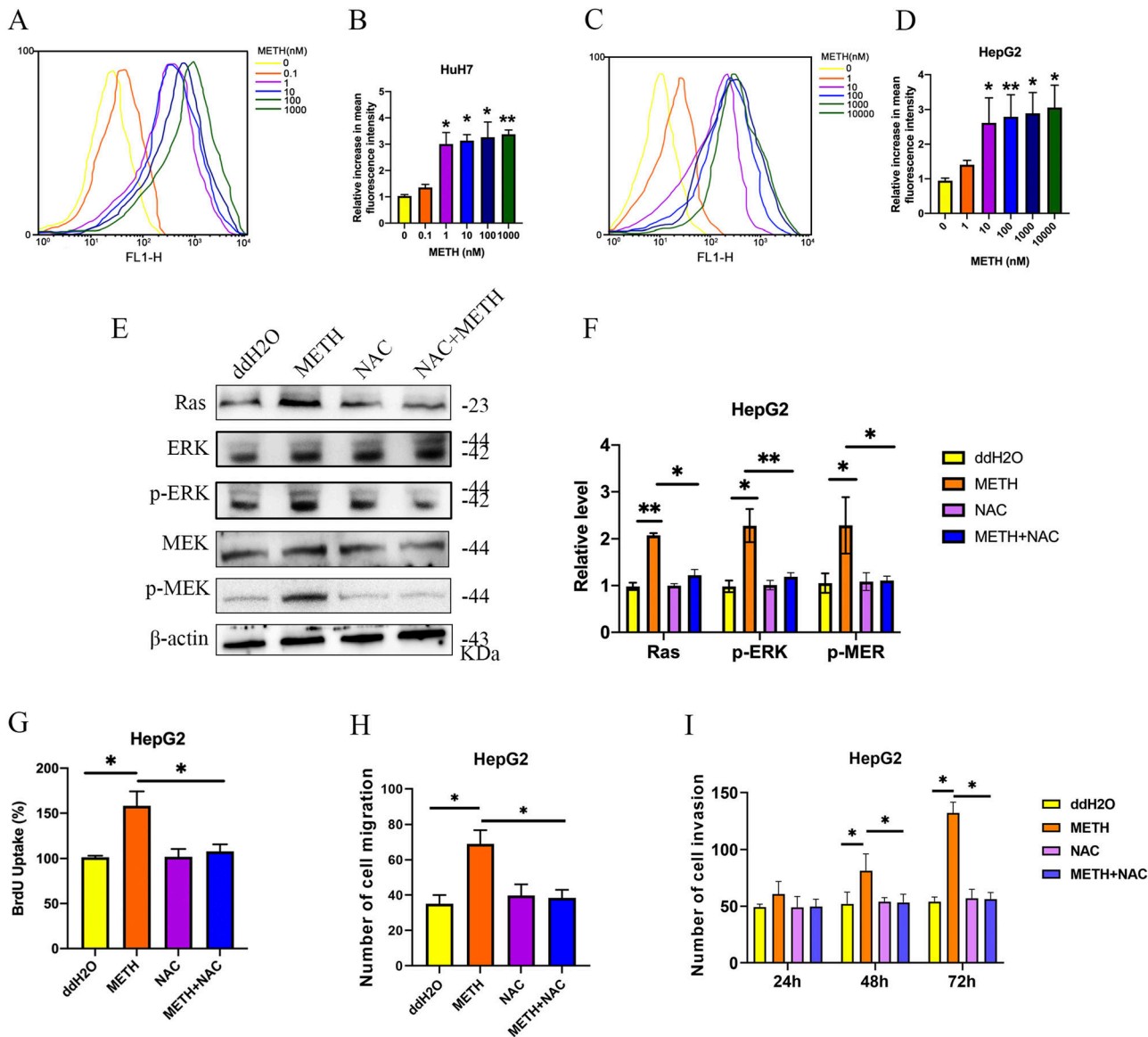

**Figure 5. METH exposure (METH/10 nM/72 h) up-regulated the ROS level, which regulated the activation of Ras/MEK/ERK pathways in cell proliferation.**
**(A, B, C, D)** Flow cytometry was used to detect the ROS levels in HUH7 (A, B) and HepG2 (C, D) cells. **(E, F)** NAC was applied to inhibit the levels of METH-induced ROS, and Western blot was used to detect the protein levels of Ras, p-MEK, MEK, p-ERK, and ERK in HepG2 (E, F) cells. **(G)** Growth rate of HepG2 cells was determined by the BrdU assay. The BrdU value in untreated cells was set as 100%. **(H, I)** Cell migration and invasion were assessed with transwell assays with mitomycin C (mean ± SD of three independent experiments). *$P < 0.05$ and **$P < 0.01$.

# Materials and Methods

## Participant selection

425 subjects were assessed for eligibility: 207 individuals were excluded, and 218 individuals completed the study (Fig 1). Among these, 103 METH users and 115 control subjects (30–60 yr old) were recruited from 2020 to 2021 (details of participants are shown in Table 1). All participants provided informed consent before the start of this study. The study was approved by the Ethics Committee of the Affiliated Hospital of Ningbo University School of Medicine.

## Drugs and reagents

METH was provided by Ningbo Public Security Bureau. N-Acetyl-L-cysteine (NAC) was purchased from Sigma-Aldrich. All drugs were dissolved in saline (0.9% NaCl) and administered to cells directly or via intraperitoneal injection. 3-(4,5-Dimethylthiazol-2-yl)-2,5-diphenyltetrazolium bromide (MTT) and DMSO were purchased from MedChemExpress. The BrdU cell proliferation kit was purchased from BioVision. The primary and secondary antibodies for Western blotting, including anti-Ras, anti-β-actin, anti-ERK, anti-pERK, anti-MEK, and anti-pMEK, were purchased from Proteintech. Polyvinylidene fluoride, blocking buffer, and ECL were

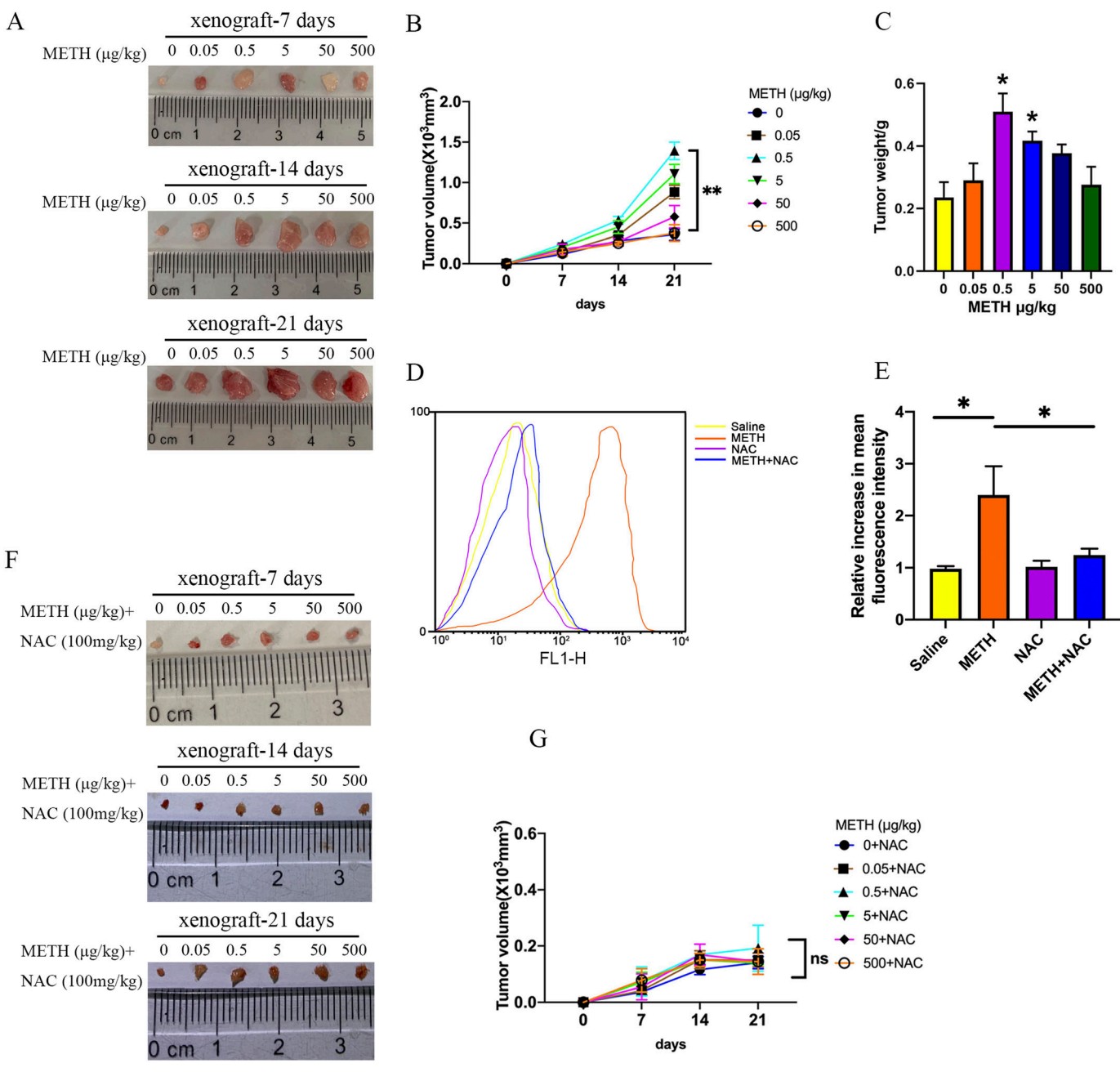

**Figure 6. METH exposure promoted xenograft tumor formation in vivo.**
HepG2 cells were injected into the immunodeficient nude mice as described in the Materials and Methods section, and mice were administered by intraperitoneal injection of various concentrations of METH. **(A, B)** Tumor volume was measured every 7 d (mean ± SEM; n = 5/group). **(C)** Tumor weight of each mouse. HepG2 cells were injected into the immunodeficient nude mice as described in the Materials and Methods section, and mice were administered by intraperitoneal injection of various concentrations of METH. **(D, E)** Flow cytometry was used to detect the ROS level in tumors. **(F, G)** NAC was applied to inhibit the level of METH-induced ROS in vivo, and tumor volume was measured every 7 d (mean ± SEM; n = 5/group). *P < 0.05 and **P < 0.01.

purchased from Millipore. The Active Ras Pull-Down and Detection Kit was purchased from Thermo Fisher Scientific.

## Animals

Male athymic BALB/c nude mice (aged 3–4 wk) were purchased from Vital River. Nude mice were housed in a standard environment (temperature, 23 ± 1°C; humidity, 40–70%) with a 12:12-h light/dark cycle. The experimental protocol was approved by the Institutional Animal Care and Use Committee of Ningbo University. All the animal experiments were conducted in accordance with the guideline for the Care and Use of Laboratory Animals published by the US National Institutes of Health.

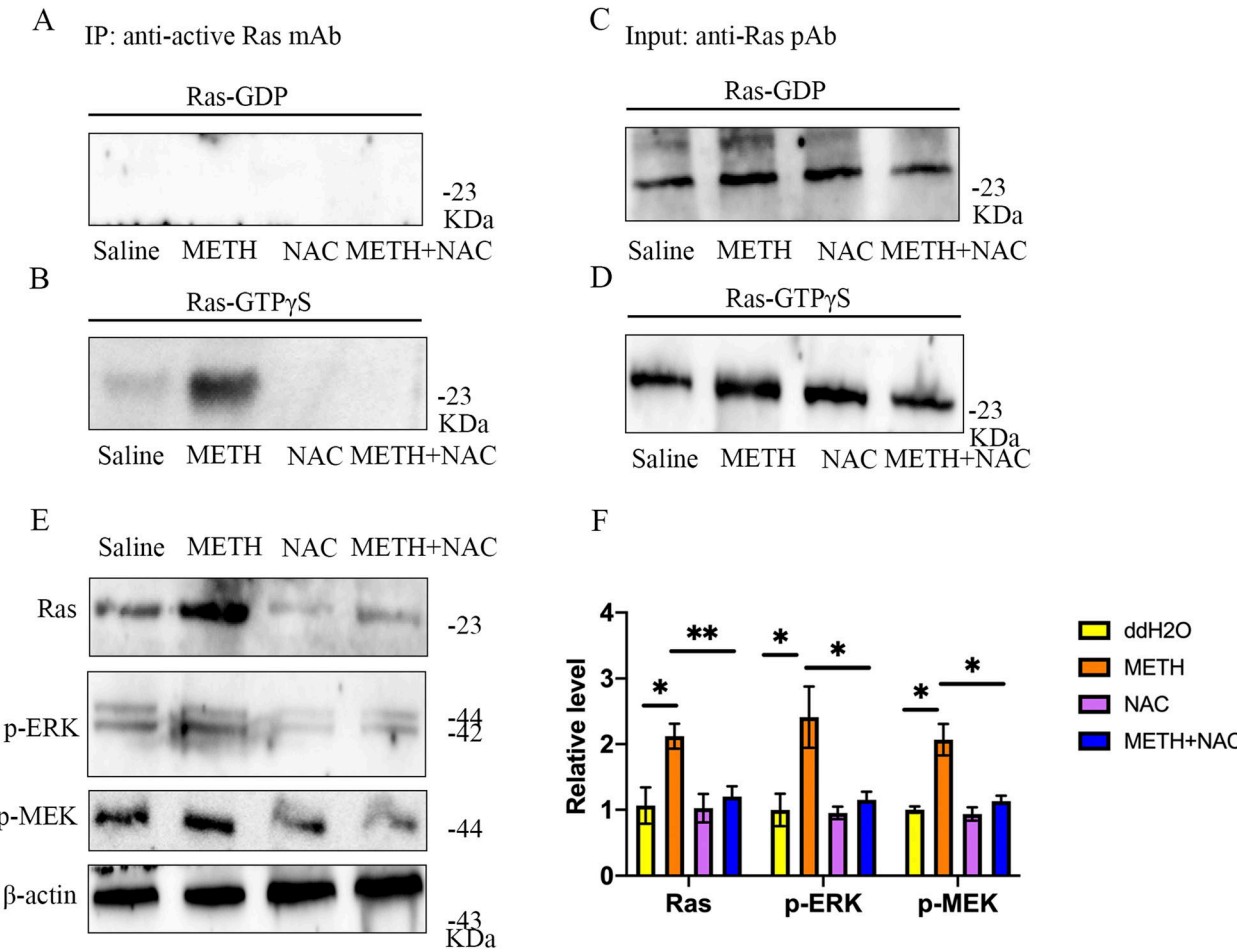

**Figure 7. ROS was involved in xenograft tumor formation and activation of Ras/MEK/ERK pathways under METH exposure.**
(A, B, C, D) Active Ras Pull-Down and Detection Kit was used to detect the activation of Ras in the tumor. (E, F) Western blot was used to detect the protein levels of Ras, p-MEK, and p-ERK in the tumor. *P < 0.05 and **P < 0.01.

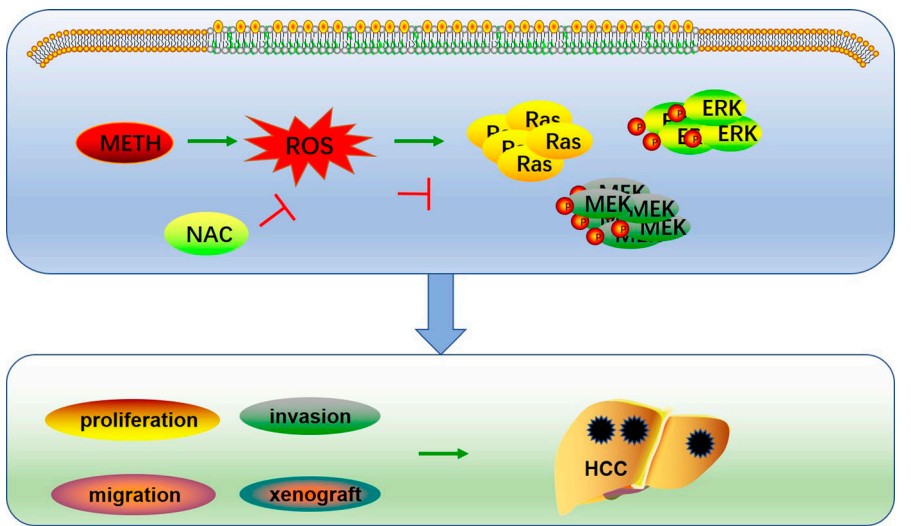

**Figure 8. ROS regulated Ras/ERK pathways under METH exposure.**
METH exposure induced ROS production, which activated the Ras/MEK/ERK signaling pathway. Clearance of ROS by NAC abolished METH-induced activation of Ras/ERK1/2 pathways leading to arrest of HCC xenograft formation in nude mice.

To assess the effects of METH on liver cancer in vivo, $6 \times 10^6$ HepG2 cells were injected subcutaneously into the right flank of nude mice. Then, mice were randomly assigned into different groups (three to five mice each) with various dosages of METH intraperitoneally injected as indicated. The mice were euthanized on days 7, 14, and 21. The xenograft tumors were isolated, photographed, and weighed. The sizes of isolated xenograft tumors were measured using a caliper. The maximum (L) and minimum (W) lengths of the tumors were measured, and the tumor size was calculated as $\frac{1}{2}LW^2$.

### Cell culture and treatment

Human liver cell lines HUH7 and HepG2 were obtained from the Cell Bank of China Science Academy. All cell lines were cultured in DMEM (Gibco) supplemented with 10% (vol/vol) FBS (Gibco), 100 U/ml penicillin, and 100 $\mu$g/ml streptomycin at 37°C and 5% $CO_2$ in a humid atmosphere. The cells used for experiments were routinely subcultured every 3 d and passaged within 10 times after thawing.

For METH treatment, HUH7 or HepG2 cells were seeded in six-well culture plates ($2 \times 10^5$/well) or 96-well culture plates ($2 \times 10^3$/well) 24 h before treatment. HUH7 cells were treated with 0, 0.1, 1, 10, 100, and 1,000 nM of METH for 72 h. HepG2 cells were treated with 0, 1, 10, 100, 1,000, and 10,000 nM of METH for 72 h. All cells were harvested and subjected to the following assays.

### MTT assay

Cell viability was detected by the MTT assay. The METH-treated cells were incubated with MTT for 4 h, and then, the solution was replaced with DMSO. Absorbance at 490 nm (OD490) was measured to determine cell viability in each group.

### BrdU assay

The BrdU assay was performed according to the manufacturer's instructions. Briefly, HUH7 cells or HepG2 cells were seeded in 96-well culture plates ($5 \times 10^3$/well) and treated with the predefined concentrations of METH for 72 h. Later, all cells were incubated with BrdU solution for 2 h at 37°C. Then, supernatants were removed, and cells were incubated with fixing/denaturing solution for 30 min at room temperature. After washing, cells were incubated with BrdU detection antibody solution for 1 h at room temperature. After 1 h of incubation with anti-mouse HRP-linked antibody solution, TMB substrate was added to each well, and the absorbance was measured at 650 nm. Then, stop solution was added to each well, and the absorbance at 450 nm was measured.

### Transwell assay

Before the transwell assay, cells were treated with 10 $\mu$g/ml mitomycin C (S8146; Selleck). For the migration assay, METH-treated cells were seeded into the upper chamber ($1 \times 10^5$/well) with 100 $\mu$l of serum-free DMEM. The lower chamber was filled with 600 $\mu$l of complete DMEM (with 10% FBS). After 24 h of culture, the non-invaded cells on the upper surface were removed, and the cells adhering to the lower surface of the filter were fixed and counted by crystal violet staining.

For the invasion assay, the membrane of the transwell unit was coated with 35 $\mu$l Matrigel at 37°C for 4 h to form a reconstructed basement membrane. Then, the cells were treated and analyzed using the same methods used for the migration assay.

### Colony formation

For the colony formation assay, the METH-treated cells were seeded in six-well plates at a density of 1,000/well. 2 wk later, cells were stained with crystal violet (0.1%), and the number of colonies was counted.

### Western blotting

Total cell or tissue extracts were extracted by cell lysis buffer followed by immunoblotting with anti-Ras (1:2,000; Proteintech), anti-$\beta$-actin (1:5,000; Proteintech), anti-ERK (1:1,000; Proteintech), anti-pERK (1:500; Proteintech), anti-MEK (1:1,000; Proteintech), and anti-pMEK (1:200; Proteintech). 25 $\mu$g of cell lysates was resolved with 12.5% SDS–PAGE and transferred to polyvinylidene fluoride. After 1-h blocking with blocking buffer, the membranes were incubated with primary antibody for 2 h at room temperature. After washing and 1-h incubation with secondary antibodies at room temperature, the blots were visualized using ECL.

### Wound healing assays

A total of $10^5$ cells/ml suspension was prepared, and 70 $\mu$l of the suspension was placed into 35-mm dishes. The serum deprivation was necessary for this assay. A 20-$\mu$l pipette tip was used to make a scratch wound. After the scratch was induced, the cells were washed with PBS and the culture medium was refreshed. Then, the cells were incubated at 37°C and 5% $CO_2$. The scratch healing area was detected after 24 h and imaged under a microscope (Olympus).

### Statistical analysis

All data were presented as the mean ± SD from at least three individual experiments. All statistical analyses were performed using GraphPad Prism 8 software. Differences in comparisons between two groups were analyzed using a $t$ test. Differences in comparisons among three or more groups were analyzed using ANOVA. Differences between METH users and control individuals were evaluated using independent $t$ tests for continuous variables and chi-squared tests for demographic and clinical variables. A $P$-value < 0.05 was statistically significant.

# Supplementary Information

## Acknowledgements

Thanks for the technical support by the Core Facilities, School of Medicine, Ningbo University. This work was supported by the National Natural Science Foundation of China (81971247), Zhejiang Provincial Key R & D Plan 2019 (2020C03064), National Natural Science Foundation of Zhejiang (LQ22H310001), Natural Science Foundation of Ningbo (2021J101), and Regular Scientific Research Project of Education Department of Zhejiang Province (Y202146346).

## Author Contributions

Z Si: formal analysis, funding acquisition, validation, visualization, methodology, and writing—original draft.
G Yang: data curation, formal analysis, validation, investigation, visualization, methodology, and writing—original draft.
X Wang: conceptualization, resources, data curation, formal analysis, investigation, visualization, and methodology.
Z Yu: software, formal analysis, validation, and investigation.
Q Pang: formal analysis, validation, visualization, and methodology.
S Zhang: investigation, visualization, and methodology.
L Qian: formal analysis, validation, and investigation.
Y Ruan: conceptualization, supervision, funding acquisition, validation, investigation, visualization, project administration, and writing—review and editing.
J Huang: conceptualization, resources, data curation, validation, investigation, visualization, and methodology.
L Yu: conceptualization, software, supervision, funding acquisition, validation, investigation, project administration, and writing—review and editing.

## Conflict of Interest Statement

The authors declare that they have no conflict of interest.

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
