## [Reviewer comments · Life Science Alliance]

Life Science Alliance

An Unconventional Cancer-Promoting function of Methamphetamine in Hepatocellular Carcinoma

Zizhen Si, GuanJun Yang, Xidi Wang, Zhaoying Yu, Qian Pang, Shuangshuang Zhang, Liyin Qian, Yuer Ruan, Jing Huang, and Liu Yu

DOI: <https://doi.org/10.26508/lsa.202201660>

Corresponding author(s): *Liu Yu, Ningbo University*

Review Timeline:

Submission Date:	2022-08-09
Editorial Decision:	2022-09-12
Revision Received:	2022-11-24
Editorial Decision:	2022-12-12
Revision Received:	2022-12-13
Accepted:	2022-12-13

Transaction Report:

September 12, 2022

Re: Life Science Alliance manuscript #LSA-2022-01660

Prof. Liu Yu
Ningbo University
818 FengHua Road Jiangbei Dist, NingBo, ZheJiang
ningbo 315211
China

Dear Dr. Yu,

Thank you for submitting your manuscript entitled "An Unconventional Cancer-Promoting function of Methamphetamine in Hepatocellular Carcinoma" to Life Science Alliance. The manuscript was assessed by expert reviewers, whose comments are appended to this letter. We invite you to submit a revised manuscript addressing the Reviewer comments.

Thank you for this interesting contribution to Life Science Alliance. We are looking forward to receiving your revised manuscript.

Sincerely,

B. MANUSCRIPT ORGANIZATION AND FORMATTING:

Reviewer #1 (Comments to the Authors (Required)):

Zizhen Sia et al manuscript "An Unconventional Cancer-Promoting function of Methamphetamine in Hepatocellular Carcinoma" showed side effect of methamphetamine to promote liver cancer. The author needs to address the following concerns.

1. Only two cancer cell lines Huh7 and HepG2 are not enough to substantiate the conclusion of METH to promote tumor. Is the 1nM concentration biologically relevant to the dose used by human?
2. Brdu proliferation did not correlate with MTT assays. Brdu showed highly significant proliferation at 10nM OR 100nm in Huh7 or HepG2 in Fig 1.? It is important to show Ki67 in western blot to substantiate the data of MTT and Brdu assays.
3. pERK is not convincing about inducible phosphorylation upon METH treatment. Phosphoprotein (pERK) is more than total protein in Fig4.

Reviewer #2 (Comments to the Authors (Required)):

Si, Yang et al., submitted the paper untitled "An unconventional cancer-promoting function of Methamphetamine in Hepatocellular carcinoma" for reviewing in Life Science Alliance.

The authors state the effect of methamphetamine (METH) in hepatocellular Carcinoma (HCC) progression. METH are a really well-known substance causing oxidative stress, extensive DNA damage and inflammatory response. Literature have reported a correlation between METH uptake and hepatic failure or cancer. This study highlights the effect of METH up-taking on HCC through ROS-mediated Ras activation.

To do so, the authors build a robust clinical cohort for their studies and correlate METH uptake with liver cancer incidence. Combination of in vitro and in vivo experiments on Huh7 and HepG2 cell line show an effect of METH on tumour growth, cell proliferation, migration and invasion through ROS-regulated Ras/MEK/ERK signalling pathway. The authors highlight this study as the first one providing evidence between METH and HCC. Methamphetamine use can therefore be included as a new risk factors for HCC.

1. Meth use was associated with a higher incidence of liver cancer.
 - The title must be written in the present tense: METH use is associate...
 - In Figure 1 and Table 1, the authors merge METH users and METH rehabilitees. Results are already significant but in order to strengthen the data and the analysis, can the authors provide METH rehabilitees data independently of METH users data?
 - Still in order to strengthen the results, the 2 non-METH users with HCC belong to the HBV/HCV subgroups, alcoholic subgroups or none?
2. METH promotes cancer-associated properties in human liver cancer cells.

General comment for Figure 2 and 3: All the experiments are performed with Huh7 and HepG2 cell line. These 2 cell lines are well-differentiated cells corresponding to early HCC stage. The authors should consider the possibility of repeating some key experiments with poorly differentiated HCC cell line corresponding to late HCC stage (by using for example HLE, HLE or SNU-449 cell line). These results can be added as supplemental figures.

- The authors base the mechanistical study on the increase of cell proliferation treated with METH. MTT assays and BrdU assays are complementary as MTT measure mitochondrial activity therefore viability and BrdU measure DNA synthesis, therefore cell proliferation. The authors need to stress the complementarity of the 2 tests in their manuscript. Viability is different

that proliferation.

- To increase the robustness of the data, apoptosis data should be added. If METH increase proliferation and viability, no apoptosis markers should be seen. The authors should check cleaved caspase 3 either by Western Blot or Immunofluorescence in the same conditions tested so far.
- Figure 2E and 2F: In addition of the colony number, the size of the colony should be measured. Representative picture of the colonies for each condition should be added.
- Figure 2G: For invasion assay, authors used transwell chamber assay with serum gradient. This experiment trigger also chemotactic behaviour. It is difficult to conclude only about invasion with this assay. In addition of the transwell assay, the authors should perform 3D invasion assay in Matrigel. The area of the spheroids will be measured overtime. Pictures of the spheroids will be added in addition of the quantification.
- Migration assays: Transwell assay is not sufficient to analyse migration properties. Wound healing assays should be performed in the same conditions. Percentage of wound recovery will be reported, as well as representative pictures.
- Most importantly, the authors show in Figure 1, an increase of the proliferative capacities of the cells treated with METH. To confirm that the effect seen in Figure 2G,H,I and J are indeed invasion and migration and not proliferation, experiments needs to be performed in presence of a proliferation inhibitor like Mitomycine C.
- Figure 2I and J representing migration capacities should be put before the invasion results.

3. METH exposure activates Ras-ERK pathways in human liver cancer cells.

- The authors show by Western Blot an upregulation of Ras after treatment of the cells with METH. The quality of some revelations needs to be increase. For example, put better exposure of Ras Figure 3C and Ras Figure 3G.
- The authors checked by Western Blot the activation of the downstream components ERK and MEK. The Ras/ERK pathway cannot be confirmed if alternative pathways are not checked. Among the several pathways that needs to be analysed, the authors should look at least by western blot: PI3K, p38, MAPkinase, ...
- The quality of some western blot needs to be increase: Fig4A: MEK (less exposed), Fig 4C P-ERK and P-MEK (less exposed).
- Figure 4B and 4D: There are too many data on the same graphic. The authors should represent the graphic with 1 graphic per each protein checked.

4. ROS regulates Ras-ERK pathway in human liver cancer cells with METH exposure.

- Figure 5E: The quality of the western blot panel needs to be increase
- Figure 5H and Figure 5I: As mentioned previously, invasion and migration should be performed in presence of an inhibitor of proliferation.

5. METH exposure promotes xenograft tumor formation through ROS-induced Ras activation in vivo.

- To validate the influence of METH on liver cancer progression, the authors injected HepG2 cells subcutaneously in nude mice. Due to the importance of tumour microenvironment in HCC and because of the sensitive parameters measured by the authors (Ros levels), the authors must in addition, inject the cells orthotopically to analyse ROS levels by Flow Cytometry.
- After Figure 6B, a figure with tumour weight should be put to confirm the observation made in Figure 6B.
- Panel of Figure 6 is difficult to follow. Some reorganisations of the panel should be done. Figure 6B should be the tumour weight of the mice presented in 6B. Not the tumour weight from mice treated with NAC.
- As the tumour growth rapidly with METH 0,5 and 5µg/kg, we expect to see fibrosis appearance. The authors should document the tumours by performing tumours Eosin/hematoxylin and Masson's trichrome colorations on tumours sections in order to identify fibrosis events.

Reviewer #1:

Zizhen Sia et al manuscript "An Unconventional Cancer-Promoting function of Methamphetamine in Hepatocellular Carcinoma" showed side effect of methamphetamine to promote liver cancer. The author needs to address the following concerns.

1. Only two cancer cell lines Huh7 and HepG2 are not enough to substantiate the conclusion of METH to promote tumor. Is the 1nM concentration biologically relevant to the dose used by human?

REPLY: Dear reviewer, thank you for the precious suggestion. We have repeated some key experiments with poorly differentiated HCC cell line corresponding to late HCC stage (by using HLE cell line) and the results were similar to that in HUH7 cells and HepG2 cells (Fig. S1). The results have been added in the Results section in **RED** as follow and 1nM concentration in cell biologically relevant to the low dose concentration used by human.

We repeated some key experiments with poorly differentiated HCC cell line corresponding to late HCC stage (by using HLE cell line) and the results were similar to that in HUH7 cells and HepG2 cells (Fig. S1).

Figure S1:

Figure S1: Impact of METH on HLE cells. (A) MTT assay was applied to determine the cell viability of HLE cells under various concentrations of METH. (B) The growth rate of HLE cells were determined by BrdU assay. The BrdU value in untreated cells was set as 100%. (C) The HLE cells were subjected to colony formation assay. (D, E) The effect of METH on hepatocellular carcinoma cell migration and invasion was assessed with transwell migration assay and transwell invasion assay, respectively. (mean \pm SD. of three independent experiments. *P < 0.05; **P < 0.01).

2. Brdu proliferation did not correlate with MTT assays. Brdu showed highly significant proliferation at 10nM OR 100nm in Huh7 or HepG2 in Fig 1.? It is important to show Ki67 in western blot to substantiate the data of MTT and Brdu assays.

REPLY: Dear reviewer, thank you for the precious suggestion. Both BrdU assay and MTT assay showed significant increases at 1nM or 10nM in Huh7 cells and 10nM or100nM in HepG2 cells. We have checked the expression of Ki67 by western blot and the result showed that METH increased the expression of Ki67 in HepG2 cells, as shown below:

Figure S2: METH increased the expression of Ki67 in HepG2 cells. (A, B) Western blot was used to detect the protein levels of Ki67 in HepG2 cell. *P < 0.05.

3.pERK is not convincing about inducible phosphorylation upon METH treatment. Phosphoprotein (pERK) is more than total protein in Fig4.

REPLY: Dear reviewer, thank you very much for your valuable suggestion. As suggested, we have put less exposed western blot in Figure 4 as follow:

Reviewer #2:

Si, Yang et al., submitted the paper untitled "An unconventional cancer-promoting function of Methamphetamine in Hepatocellular carcinoma" for reviewing in Life Science Alliance.

The authors state the effect of methamphetamine (METH) in hepatocellular Carcinoma (HCC) progression. METH are a really well-known substance causing oxidative stress, extensive DNA damage and inflammatory response. Literature have reported a correlation between METH uptake and hepatic failure or cancer. This study highlights the effect of METH up-taking on HCC through ROS-mediated Ras activation.

To do so, the authors build a robust clinical cohort for their studies and correlate METH uptake with liver cancer incidence. Combination of in vitro and in vivo experiments on Huh7 and HepG2 cell line show an effect of METH on tumour growth, cell proliferation, migration and invasion through ROS-regulated Ras/MEK/ERK signalling pathway. The authors highlight this study as the first one providing evidence between METH and HCC. Methamphetamine use can therefore be included as a new risk factors for HCC.

1. Meth use was associated with a higher incidence of liver cancer.

- The title must be written in the present tense: METH use is associate...

REPLY: Dear reviewer, thank you for the precious suggestion. As suggested, we have corrected the title in the Results section in **RED** as follow:

3.1 METH use is associated with a higher incidence of liver cancer

- In Figure 1 and Table 1, the authors merge METH users and METH rehabilitees. Results are already significant but in order to strengthen the data and the analysis, can the authors provide METH rehabilitees data independently of METH users data?

REPLY: Dear reviewer, thank you for the precious suggestion. As suggested, we have provided METH rehabilitees data independently of METH users data in the Supplementary Materials section in **RED** as follow:

A total of 218 participants were included. Among these, 103 were diagnosed as METH users or METH rehabilitees (Table S1).

METH rehabilitees			
Total sample (n=87)		HCC (n=3)	Without HCC (n=84)
HBV positive		2	3
HCV positive		1	2
Alcohol use		3	56
Sex	Male	2	51
	Female	1	32
Age(years)	30-39	0	33
	40-49	2	31
	50-60	1	20

Table S1: Demographics of METH rehabilitees.

- Still in order to strengthen the results, the 2 non-METH users with HCC belong to the HBV/HCV subgroups, alcoholic subgroups or none?

REPLY: Dear reviewer, thank you for the precious suggestion. As suggested, we have provided non-METH users data in the Supplementary Materials section in **RED** as follow and the results showed that 1 non-METH users with HCC belong to the HBV subgroups, and 2 non-METH users belong to alcoholic subgroups.

A total of 218 participants were included. Among these, 103 were diagnosed as METH users or METH rehabilitees (Table S1), and 115 were age and gender-matched normal subjects with no history of METH abuse (Table S2).

Non-METH users		
Total sample (n=115)	HCC (n=2)	Without HCC (n=113)
HBV positive	1	5
HCV positive	0	3
Alcohol use	2	60
Sex	Male	41
	Female	72
Age(years)	30-39	51
	40-49	42
	50-60	20

Table S2: Demographics of Non-METH users.

2. METH promotes cancer-associated properties in human liver cancer cells.

General comment for Figure 2 and 3: All the experiments are performed with Huh7 and HepG2 cell line. These 2 cell lines are well-differentiated cells corresponding to early HCC stage. The authors should consider the possibility of repeating some key experiments with poorly differentiated HCC cell line corresponding to late HCC stage (by using for example HLE, HLE or SNU-449 cell line). These results can be added as supplemental figures.

REPLY: Dear reviewer, thank you for the precious suggestion. We have repeated some key experiments with poorly differentiated HCC cell line corresponding to late HCC stage (by using HLE cell line) and the results were similar to that in HUH7 cells and HepG2 cells (Fig. S1). The results have been added in the Results section in RED as follow and 1nM concentration in cell biologically relevant to the low dose concentration used by human.

We repeated some key experiments with poorly differentiated HCC cell line corresponding to late HCC stage (by using HLE cell line) and the results were similar to that in HUH7 cells and HepG2 cells (Fig. S1).

Figure S1:

Figure S1: Impact of METH on HLE cells. (A) MTT assay was applied to determine the cell viability of HLE cells under various concentrations of METH. (B) The growth rate of HLE cells were determined by BrdU assay. The BrdU value in untreated cells was set as 100%. (C) The HLE cells were subjected to colony formation assay. (D, E) The effect of METH on hepatocellular carcinoma cell migration and invasion was assessed with transwell migration assay and transwell invasion assay, respectively. (mean \pm SD. of three independent experiments. * $P < 0.05$; ** $P < 0.01$).

- The authors base the mechanistical study on the increase of cell proliferation treated with METH. MTT assays and BrdU assays are complementary as MTT measure mitochondrial activity therefore viability and BrdU measure DNA synthesis, therefore cell proliferation. The authors need to stress the complementarity of the 2 tests in their manuscript. Viability is different that proliferation.

REPLY: Dear reviewer, thank you for the precious suggestion. We have modified Result 3.2 to address this issue in **RED**, as shown below:

Then, the cell viability was determined by MTT assays and cell proliferation ability was determined by BrdU assays. The MTT assay showed that treatment with low concentrations of

METH significantly promoted cell viability in HUH7 cells (0.1, 1, or 10 nM) and HepG2 cells (1, 10 or 100nM) (Fig. 2A, B). Then, HUH7 or HepG2 cells were exposed to METH for 72 hours and underwent the BrdU assay. As shown in Fig. 1C, D, low concentrations of METH promoted cell proliferation in HUH7 cells (0.1, 1, or 10 nM) and HepG2 cells (1, 10 or 100nM) (Fig. 2C, D).

- To increase the robustness of the data, apoptosis data should be added. If METH increase proliferation and viability, no apoptosis markers should be seen. The authors should check cleaved caspase 3 either by Western Blot or Immunofluorescence in the same conditions tested so far.

REPLY: Dear reviewer, thank you for the precious suggestion. We have checked the expression of cleaved caspase 3 by western blot and the result showed that METH decreased the expression of cleaved caspase 3 in HepG2 cells, as shown below:

Additionally, cell apoptosis was determined by western blot and METH decreased the expression of cleaved caspase 3 (Fig. S2 C, D).

Figure S2:

Figure S2: METH decreased the expression of cleaved caspase 3 in HepG2 cell. (C, D)

Western blot was used to detect the protein levels of cleaved caspase 3 in HepG2 cell. *P < 0.05.

- Figure 2E and 2F: In addition of the colony number, the size of the colony should be measured. Representative picture of the colonies for each condition should be added.

REPLY: Dear reviewer, thank you for the precious suggestion. Clonogenic assay or colony formation assay is an in vitro cell survival assay based on the ability of a single cell to grow into a colony. The colony size and representative picture for each condition have been added in the Results section in RED according to your suggestion as follow:

Consistently, the colony formation assay showed that HUH7 cells and HepG2 cells treated with

low concentrations of METH could form more colonies compared to the ddH2O control group (Fig. 2E, F; Fig. S3A-D).

Figure S3:

Figure S3: The colony size and representative picture for each group. (A, B) Colony size in pixels for different groups. *P < 0.05. (C, D) Representative images of colonies.

- Figure 2G: For invasion assay, authors used transwell chamber assay with serum gradient. This experiment trigger also chemotaxic behaviour. It is difficult to conclude only about invasion with this assay. In addition of the transwell assay, the authors should perform 3D invasion assay in Matrigel. The area of the spheroids will be measured overtime. Pictures of the spheroids will be added in addition of the quantification.

REPLY: Dear reviewer, thank you very much for your valuable suggestion. We also carefully

considered the chemotaxis of the HUH7 and HepG2 cells when implementing the invasion assay. To avoid this, for each condition the media in upper chamber and lower chamber both had same concentration of METH. In this situation, the METH gradient did not establish and chemotactic behaviour was not induced. Thus, we believe that more invasive cells observed in METH groups compared to control groups was resulted from enhanced invasion capabilities stimulated by METH.

- Migration assays: Transwell assay is not sufficient to analyse migration properties. Wound healing assays should be performed in the same conditions. Percentage of wound recovery will be reported, as well as representative pictures.

REPLY: Dear reviewer, thank you very much for your constructive comments to strengthen this manuscript. As suggested, wound healing assays were performed in the same conditions. The migration speed and representative picture for each condition have been added in the Results section in RED as follow:

Further, the wound healing assay also showed the same result (Fig. S4 A-D).

Figure S4:

Figure S4: Wound healing assays were performed for each group. (A, B) Representative

images of wound healing assays. (C, D) Migration speed for different groups. *P < 0.05.

- Most importantly, the authors show in Figure 1, an increase of the proliferative capacities of the cells treated with METH. To confirm that the effect seen in Figure 2G,H,I and J are indeed invasion and migration and not proliferation, experiments needs to be performed in presence of a proliferation inhibitor like Mitomycin C.

REPLY: Dear reviewer, thank you very much for your constructive comments to strengthen this manuscript. As suggested, 10 µg/mL Mitomycin C was applied to the cells and the results have been added in the Results section in **RED** as follow:

Next, the invasion and migration ability of the two types of cells (with 10 µg/mL Mitomycin C) were determined by trans-well assay with or without Matrigel. Similar to the effects of METH on proliferation potential, low concentrations of METH treatment promoted the migration and invasion ability of HUH7 and HepG2 cells (Fig. 2G-J).

Revised figure 2:

Figure 2: Impact of METH on HUH7 and HepG2 cells. (A, B) MTT assay was applied to

determine the cell viability of HUH7 and HepG2 cells under various concentrations of METH. (C, D) The growth rate of HUH7 and HepG2 cells were determined by BrdU assay. The BrdU value in untreated cells was set as 100%. (E, F) The HUH7 and HepG2 cells were subjected to colony formation assay. (G-J) The effect of METH on hepatocellular carcinoma cell migration and invasion was assessed with transwell migration assay and transwell invasion assay, respectively. (mean \pm SD. of three independent experiments. *P < 0.05; **P < 0.01).

- Figure 2I and J representing migration capacities should be put before the invasion results.

REPLY: Dear reviewer, thank you very much for your valuable suggestion. As suggested, the migration results have been put before the invasion results in Figure 2 as follow:

Figure 2:

Figure 2: Impact of METH on HUH7 and HepG2 cells. (A, B) MTT assay was applied to determine the cell viability of HUH7 and HepG2 cells under various concentrations of METH. (C, D) The growth rate of HUH7 and HepG2 cells were determined by BrdU assay. The BrdU value

in untreated cells was set as 100%. (E, F) The HUH7 and HepG2 cells were subjected to colony formation assay. (G-J) The effect of METH on hepatocellular carcinoma cell migration and invasion was assessed with transwell migration assay and transwell invasion assay, respectively. (mean \pm SD, of three independent experiments. *P < 0.05; **P < 0.01).

3. METH exposure activates Ras-ERK pathways in human liver cancer cells.

- The authors show by Western Blot an upregulation of Ras after treatment of the cells with METH. The quality of some revelations needs to be increase. For example, put better exposure of Ras Figure 3C and Ras Figure 3G.

REPLY: Dear reviewer, thank you very much for your valuable suggestion. As suggested, we have put better exposure of Ras in Figure 3 as follow:

- The authors checked by Western Blot the activation of the downstream components ERK and MEK. The Ras/ERK pathway cannot be confirmed if alternative pathways are not checked. Among the several pathways that needs to be analysed, the authors should look at least by western blot: PI3K, p38, MAPkinase, ...

REPLY: Dear reviewer, thank you very much for your valuable suggestion. As suggested, we have checked the expression of these proteins. We found that METH increased the expression of PI3K and MAPK in HUH7 cells and HepG2 cells. These results have been put in Results section in RED as follows:

(I, J) Western blot was used to detect the protein levels of PI3K and MAPK under 1nM METH exposure for 72h in HUH7 cells. (K, L) Western blot was used to detect the protein levels of PI3K and MAPK under 10nM METH exposure for 72h in HepG2 cells. (mean \pm SD. of three independent experiments. *P < 0.05; **P < 0.01).

- The quality of some western blot needs to be increase: Fig4A: MEK (less exposed), Fig 4C P-ERK and P-MEK (less exposed).

REPLY: Dear reviewer, thank you very much for your valuable suggestion. As suggested, we have put less exposed western blot in Figure 4 as follow:

- Figure 4B and 4D: There are too many data on the same graphic. The authors should represent the graphic with 1 graphic per each protein checked.

REPLY: Dear reviewer, thank you very much for your valuable suggestion. As suggested, we

have represented the graphic with 1 graphic per each protein checked in Figure 4 as follow:

4. ROS regulates Ras-ERK pathway in human liver cancer cells with METH exposure.

- Figure 5E: The quality of the western blot panel needs to be increase

REPLY: Dear reviewer, thank you very much for your valuable suggestion. As suggested, we have put better western blot panels in Figure 5 as follow:

- Figure 5H and Figure 5I: As mentioned previously, invasion and migration should be performed in presence of an inhibitor of proliferation.

REPLY: Dear reviewer, thank you very much for your constructive comments to strengthen this

manuscript. As suggested, 10 µg/mL Mitomycin C was applied to the cells and the results have been added in the Results section in RED as follow:

Figure 5. (H, I) Cell migration and invasion were assessed with transwell assays with Mitomycin C. (mean ± SD. of three independent experiments. *P < 0.05; **P < 0.01).

5. METH exposure promotes xenograft tumor formation through ROS-induced Ras activation in vivo.

- To validate the influence of METH on liver cancer progression, the authors injected HepG2 cells subcutaneously in nude mice. Due to the importance of tumour microenvironment in HCC and because of the sensitive parameters measured by the authors (Ros levels), the authors must in addition, inject the cells orthotopically to analyse ROS levels by Flow Cytometry.

REPLY: Dear reviewer, thank you very much for your valuable suggestions. As a powerful ROS inducer, METH promotes ROS production in almost all cell types. In our study, we focused on the effects of METH on ROS production in HCC tumor cells. Our in vitro experiments demonstrated that addition of METH significantly elevated ROS level in two different HCC cell lines. Consistently, our in vivo study found that METH administration remarkably increased ROS level in tumor xenografts compared to control group. These findings strongly supported that METH promoted ROS production in HCC tumor cells. Similar experiments designs were performed in many other studies (Ref: PMID: 35153295, 33875663, 32929349). Your valuable suggestions give us a promising direction for our future study of the role of METH on tumor microenvironment. We appreciate for that!

- After Figure 6B, a figure with tumor weight should be put to confirm the observation made in Figure 6B.

REPLY: Dear reviewer, thank you very much for your constructive comments to strengthen this manuscript. As suggested, a figure with tumor weight has been put to confirm the observation made in Figure 6B as follow.

Figure 6. (B) Tumor volume was measured every 7 days (mean \pm SEM. n =5/group). (C) Tumor weight of each mouse. HepG2 cells were injected into the immunodeficient nude mice as described in methods, and mice were administrated by intraperitoneal injection of various concentrations of METH.

- Panel of Figure 6 is difficult to follow. Some reorganisations of the panel should be done. Figure 6B should be the tumour weight of the mice presented in 6B. Not the tumour weight from mice treated with NAC.

REPLY: Dear reviewer, thank you very much for your constructive comments to strengthen this manuscript. As suggested, we have reorganized Figure 6 as follows.

Figure 6: METH exposure promoted xenograft tumors formation in vivo. HepG2 cells were injected into the immunodeficient nude mice as described in methods, and mice were administered with intraperitoneal injections of various concentrations of METH. (A, B) Tumor volume was measured every 7 days (mean \pm SEM. n =5/group). (C) Tumor weight of each mouse. HepG2 cells were injected into the immunodeficient nude mice as described in methods, and mice were administrated by intraperitoneal injection of various concentrations of METH. (D, E) Flow Cytometry was used to detect the ROS level in tumors. (F, G) NAC was applied to inhibit the level of METH induced-ROS in vivo, and tumor volume was measured every 7 days (mean \pm SEM. n =5/group). *P < 0.05; **P < 0.01.

- As the tumour growth rapidly with METH 0,5 and 5µg/kg, we expect to see fibrosis appearance. The authors should document the tumours by performing tumours Eosin/hematoxylin and Masson's trichrome colorations on tumours sections in order to identify fibrosis events.

REPLY: Dear reviewer, thank you very much for your valuable suggestions. Fibrosis can happen in a rapidly growing tumor. In our current study, we focused on the effects of METH on tumor

growth. As shown in Fig. 6A, compared to control group, the tumor xenografts grew faster upon METH administration and the sizes of tumor xenografts in mice of METH administration group were larger. These findings strongly supported our hypothesis that METH promoted HCC growth *in vivo*.

December 12, 2022

RE: Life Science Alliance Manuscript #LSA-2022-01660R

Prof. Liu Yu
Ningbo University
818 FengHua Road Jiangbei Dist, NingBo, ZheJiang
ningbo 315211
China

Dear Dr. Yu,

Thank you for submitting your revised manuscript entitled "An Unconventional Cancer-Promoting function of Methamphetamine in Hepatocellular Carcinoma". We would be happy to publish your paper in Life Science Alliance pending final revisions necessary to meet our formatting guidelines.

- please upload your main and supplementary figures as single files; please upload your table files as separate editable doc or excel files, or make sure that they are in the doc file of your main manuscript
- please add the Twitter handle of your host institute/organization as well as your own or/and one of the authors in our system
- please consult our manuscript preparation guidelines <https://www.life-science-alliance.org/manuscript-prep> and make sure your manuscript sections are in the correct order
- please use the [10 author names, et al.] format in your references (i.e. limit the author names to the first 10)
- please double-check your callouts for Figure 1; you have a callout for panel C and D, but these are not in the figure or the figure legend
- please add a callout for Figure 4H and Figure S2 A,B to your main manuscript text
- please add sizes next to all blots

Figure Check:

-In Figure 4E, ERK row and Figure 4K, PI3K row: there appears to be a horizontal line through these rows. If this was added, please remove. If this is an artifact, please let us know in your Cover Letter.

A. FINAL FILES:

B. MANUSCRIPT ORGANIZATION AND FORMATTING:

Sincerely,

Reviewer #1 (Comments to the Authors (Required)):

Author addressed all the concerns.

Reviewer #2 (Comments to the Authors (Required)):

The authors Sia, Yang et al. submitted the manuscript untitled "An unconventional cancer-promoting function of Methamphetamine in Hepatocellular Carcinoma" after performing the revisions asked by the reviewers.

In general, the authors have correctly answered the questions and additional experiments that were requested. The quality of the manuscript is now enhanced compared to the first submission. Nevertheless, the authors could have taken more time and performed the requested orthotopic injections. It is difficult to convince the readers of the effect of METH in tumour progression if the in-vivo experiments are not more extensive. In case the technique is not established in the lab, a collaboration could have been established with an expert research team. The data obtained after orthotopic injection of the cells and flow cytometric analysis of the ROS would have considerably reinforced the message of the paper.

-please upload your main and supplementary figures as single files; please upload your table files as separate editable doc or excel files, or make sure that they are in the doc file of your main manuscript.

REPLY: Dear editor, thank you for the precious suggestion. We have uploaded our main and supplementary figures as single files and uploaded table files as separate editable excel files.

-please add the Twitter handle of your host institute/organization as well as your own or/and one of the authors in our system

REPLY: Dear editor, thank you for the precious suggestion. We are sorry that our institute/organization do not have Twitter account.

-please consult our manuscript preparation guidelines <https://www.life-science-alliance.org/manuscript-prep> and make sure your manuscript sections are in the correct order

REPLY: Dear editor, thank you for the precious suggestion. We have consulted the preparation guidelines and make sure our manuscript sections are in the correct order.

-please use the [10 author names, et al.] format in your references (i.e. limit the author names to the first 10)

REPLY: Dear editor, thank you for the precious suggestion. We have limited the author names to the first 10 in the references.

-please double-check your callouts for Figure 1; you have a callout for panel C and D, but these are not in the figure or the figure legend

REPLY: Dear editor, thank you for the precious suggestion. We have checked the callouts for Figure 1 and corrected the callout for panel C and D in **RED**.

-please add a callout for Figure 4H and Figure S2 A,B to your main manuscript text

REPLY: Dear editor, thank you for the precious suggestion. We have checked the callouts for Figure 4H and Figure S2 A,B in the main manuscript text in **RED**.

-please add sizes next to all blots

REPLY: Dear editor, thank you for the precious suggestion. We have added sizes next to all blots.

Figure Check:

-In Figure 4E, ERK row and Figure 4K, PI3K row: there appears to be a horizontal line through these rows. If this was added, please remove. If this is an artifact, please let us know in your Cover Letter.

REPLY: Dear editor, thank you for the precious suggestion. We have removed the lines.

REPLY: Dear editor, thank you for the precious suggestion. We will.

December 13, 2022

RE: Life Science Alliance Manuscript #LSA-2022-01660RR

Prof. Liu Yu
Ningbo University
818 FengHua Road Jiangbei Dist, NingBo, ZheJiang
NingBo 315211
China

Dear Dr. Yu,

Thank you for submitting your Research Article entitled "An Unconventional Cancer-Promoting function of Methamphetamine in Hepatocellular Carcinoma". It is a pleasure to let you know that your manuscript is now accepted for publication in Life Science Alliance. Congratulations on this interesting work.

DISTRIBUTION OF MATERIALS:

Again, congratulations on a very nice paper. I hope you found the review process to be constructive and are pleased with how the manuscript was handled editorially. We look forward to future exciting submissions from your lab.

Sincerely,
